# Health improvement of the elderly in five Central Asian countries during COVID-19 based on difference game

Yuntao Bai[1], Lan Wang[2]*, Shuang Xu[1]

1 Business School, Shandong Management University, Jinan, China, 2 Center of Emergency Management, Chongqing Academy of Governance, Chongqing, China

* wanglan-8722@hotmail.com

**Data Availability Statement:** All relevant data are within the paper and its Supporting Information files.

## Abstract

In 2020, COVID-19 became a global pandemic. Older people are less resistant to the novel coronavirus. In order to ensure the health of the elderly population, the governments of five Central Asian countries should provide home medical services for the elderly or provide "green channel" to medical services. This "green channel" means providing a special service and treatment for the elderly in the hospital to ensure that they can safely and easily access the medical services they need. In order to study the application scope of various modes, this article constructs three modes of differential game: no special care, home medical care and "green channel". And the equilibrium results are compared and analyzed. Research shows that when the additional medical costs associated with house calls or "green channel" gradually increase, the social benefits to both the elderly and the government gradually decrease, and eventually it is less than the social benefits under the no-special care model. The greater the credibility of the government under the "green channel" service model, the greater the social benefits of the government. However, the greater the credibility of the government under the home medical service model, the smaller the social benefits of the government.

## 1. Introduction

### 1.1 Background and research significance

The novel coronavirus epidemic is a major health emergency with the fastest spreading speed, the most widespread infection and the most difficult prevention and control in recent decades. It has had a huge impact on public health systems in many countries around the world. Compared with developed countries such as Europe and America, the medical and health system of five Central Asian countries is relatively backward [1]. At the same time, five Central Asian countries have older populations. Take five Central Asian Kazakhstan as an example. According to UN statistics, by the end of 2020, residents aged 60 and over 65 will account for 11.6 percent and 7.5 percent of Kazakhstan's population, respectively [2]. This means that Kazakhstan has entered an aging society. Older people have lower resistance and higher mortality rates

**Funding:** This work is financially supported by Social Science Planning Foundation of Chongqing in China (2021BS080); This work is financially supported by National Natural Science Foundation of China (72304157). The funders had no role in study design, data collection and analysis, decision to publish, or preparation of the manuscript.

**Competing interests:** The authors have declared that no competing interests exist.

than younger people [3]. The aging of five Central Asian countries has made it more difficult for them to cope with COVID-19. Under the limited medical infrastructure, how to provide more effective medical services for the elderly and reduce the morbidity and mortality of the elderly is an important issue.

Many older people have underlying medical conditions that increase their risk of mortality from COVID-19 infection [4]. Introducing advanced medical devices, vaccinations and purchasing specific drugs are important measures to effectively control the COVID-19 epidemic and reduce morbidity and mortality among the elderly. However, the level of economic development of five Central Asian countries has more or less appeared certain difficulties. It is difficult for these countries to guarantee the life and health of the elderly by improving the level of medical equipment and drugs. For this reason, in order to protect the life, health and safety of the elderly, it is difficult for five Central Asian countries to rely solely on equipment support, technical means and funds. These countries will have to make some administrative changes to their existing healthcare systems [5].

In order to achieve the health level of the elderly in five Central Asian countries, some management measures can be adopted, such as providing home medical services for the elderly, or providing "green channel" to medical services for the elderly. The so-called home medical service means that when the elderly show certain symptoms, medical workers go to the elderly's home for medical services. This is mainly due to the mobility of the elderly themselves, as well as to prevent cross-infection. But this approach costs more and does not allow older people to receive more specialized care. The green medical channel means that the elderly go to the hospital for medical treatment when they are unwell, and at the same time, the hospital opens a "green channel" for the elderly. The "green channel" opened in hospitals during the COVID-19 pandemic is to ensure that the elderly have timely access to medical services and care. Especially the elderly, who are more vulnerable and susceptible to the novel coronavirus. Specifically, the "green channel" usually has priority arrangements and visits, convenient procedures for seeing a doctor, dedicated service personnel, designated medical institutions, special guarantees and arrangements. In summary, the "green channel" aims to ensure that the elderly enjoy priority rights in the medical treatment process, reducing waiting time and inconvenience. This can provide them with more humane and intimate medical services to protect their health and safety. Although this can enable the elderly to receive more professional and authoritative treatment, it is easy to cause cross infection among the elderly.

## 1.2 Literature review

The novel coronavirus pandemic has swept the world, causing a huge impact on people's health. Among them, the physical health of the elderly is affected more. For example, if you compare the mortality rates of older adults aged 50–64, 65–79, and over 80 years, you will find that the higher the age, the higher the mortality rate. And, overall, the mortality rate of older adults reached 8.3% [6]. At the same time, compared with younger people, it was found that older adults were more affected, with about half of older COVID-19 patients having severe infections, one fifth critically ill, and one tenth dying [7]. Some scholars have studied the impact of COVID-19 on the elderly. Firstly, on the psychological level, COVID-19 can cause anxiety and depression in the elderly [8,9], and at the same time, the social loneliness and stress perception of the elderly increase [10]. Secondly, in terms of hospitalization rate, COVID-19 leads to an increase in the hospitalization rate of the elderly [11]. Finally, in terms of mortality rate, COVID-19 can cause a higher mortality rate in the elderly [3], and this phenomenon is more obvious for the elderly with cardiovascular disease [4].

The vaccination rate, the supply of medical resources and the medical system play a very important role in the prevention and control of the epidemic in five Central Asian countries. Some scholars have analyzed medical conditions in five Central Asian countries. Specifically, they include increased drug resistance [12], increased prevalence and vaccination rates of infectious diseases [13], reduced accessibility and affordability [14], and backwardness of the health care system [15]. However, these scholars did not study the specific medical models in Central Asian countries and the effects of different medical models on the physical health of the elderly.

Prevention and treatment modalities for older persons are more complex due to the physical characteristics of older persons [16]. To address these concerns, some scholars have looked at how to mitigate the impact of COVID-19 on the elderly. For example, the safety and efficacy of the Sinopharmate vaccine (BBIBP-CorV) in the elderly population was analyzed in Faisalabad, Pakistan [17]. Vaccination can effectively reduce lung disease caused by COVID-19 [18]. Efficacy of specific drugs such as Remdesivir in elderly patients with COVID-19 [19]. However, these scholars analyzed how to safeguard the health of the elderly under the epidemic more from the technical level, rather than from the management perspective.

In order to make up for the shortcomings of the above research, this article studies the health problems of the elderly in five Central Asian countries from the perspective of various medical service modes. This article divides medical service mode into home medical service and "green channel" mode. The equilibrium results under different medical service modes were compared and analyzed. Finally, the scope of application of various medical service modes is obtained. It provides reference for the reform of medical system in five Central Asian countries.

Meanwhile, this paper uses differential game as the research method. Differential game refers to a time continuous game played by multiple players in a time continuous system. It has the goal of optimizing the independence and conflict of each player, and can finally obtain the strategy of each player evolving over time and reach the Nash equilibrium. Considering that the medical service system in the five Central Asian countries is a dynamic and diversified complex system, which has multiple players such as governments, medical institutions, patients, etc., which have competitive and cooperative relations with each other. By incorporating these factors into the framework of differential game, we can analyze the advantages, disadvantages and potential improvement measures of different medical service models under different conditions. At present, the differential game it is mainly applied in the fields of environmental protection [20], pricing strategy [21] and advertising decision [22]. The novel coronavirus is constantly mutating. In addition, the infection rate, hospitalization rate and death rate of the elderly due to COVID-19 are constantly changing. Five Central Asian countries' epidemic prevention and control policies are also changing. Therefore, this article uses differential game, a time continuous game method.

## 2. Methodology

An ethical statement is made first before starting the method. The research of this paper is mainly based on the differential game model, comparing and analyzing the different models, and finally drawing the research conclusion. This article does not refer to any specific individuals, institutions, or specific research data, and there are no potential conflicts of interest that may exist.

### 2.1 Problem description, hypothesis, and variable definition

**2.1.1 Problem description.** As the world opens up to the COVID-19 pandemic, the plight of the five Central Asian elderly is being laid bare. This is mainly caused by two reasons. On

the one hand, due to the physiological aging of individuals and the decline of immunity, the elderly become the first vulnerable group during the epidemic [23]. On the other hand, the huge number of people seeking medical treatment due to the COVID-19 pandemic has crowded out medical resources for the elderly. Therefore, society should give special care to the elderly. For example, home medical services for the elderly or "green channel" in hospitals for the elderly to see a doctor. Although providing medical services to the elderly can avoid reinfection or cross-infection of the elderly, the cost is high. Although providing the elderly with "green channel" to medical services in hospitals is cheaper and enables them to receive better medical treatment than at home, it is easy for the elderly to be reinfected with the novel coronavirus.

During the COVID-19 pandemic, the health care resources provided to the elderly population in the five Central Asian countries are "dynamic". This is mainly caused by the following reasons. First, the change of the epidemic. The COVID-19 epidemic is constantly changing on a global scale. The five Central Asian countries are differently affected by the epidemic, and there are significant differences in the resource demands corresponding to the peak and trough periods. Therefore, with the change of the epidemic, the demand for health care resources for the elderly will also be adjusted accordingly, and the dynamic allocation will be realized [24]. Second, the imbalance of medical resources. There is an obvious imbalance of medical resources in different regions and different countries. When the epidemic breaks out, the regions with rich resources may strengthen the support to the regions with limited resources, and realize the effective use of resources through dynamic allocation to meet the needs of the epidemic. Third, the emergency needs. The COVID-19 epidemic is an emergency public health event. Many areas of the health care system in the five Central Asian countries may not be fully prepared at the early stage, and face the shortage of urgently needed materials such as medical masks, protective suits, and ventilators. In this case, the allocation of health care resources for the elderly in the five Central Asian countries must be flexible to quickly meet the emergency needs. Fourth, the progress of vaccination. With the success and popularity of vaccine development, the vaccination schedule in different regions of the five countries in Central Asia is also different, which affects the control effect and the number of cases. The allocation of health care resources needs to be adjusted accordingly according to the change in the vaccination schedule, so as to allocate resources reasonably. Fifth, priority adjustment. As the epidemic evolves, some urgent and major cases may need to be given priority for medical treatment [25]. This means that governments and medical institutions need to constantly adjust their priorities to ensure that resources are fully utilized in the places that need the most attention. Sixth, information update. As researchers continue to make new discoveries in the study of COVID-19, the methods of patient diagnosis and treatment and epidemic prevention measures are constantly updated. The adjustment and allocation of resources based on the latest information can better meet the challenges of the epidemic. Therefore, during the COVID-19 pandemic, the allocation of health care resources needs to be flexibly adjusted according to the actual situation to achieve dynamic optimization. This helps to control the epidemic more effectively, reduce the infection rate and increase the recovery rate.

The game parties in this article are mainly the elderly and the government. In order to provide better medical services for the elderly, the government can adopt the following three medical service models:

1. Medical service mode without special care. What's causing the current frenzy is the much-talked about variant of Omicron. Omicrone is a surprise, with strong infectivity, a lack of specific symptoms after infection, survival on plastic surfaces for eight days, and the new subtype variant "evolved" again. Because the symptoms of Omicron infection are not

obvious in most patients, many people have no symptoms. Some are mainly fever, dry cough, headache, nasal congestion, fatigue, sore throat and other lack of specific symptoms, resulting in the transmission of the occultness is very strong, more prone to multiple sporadic or concentrated outbreaks. Some areas are under great financial pressure and lack of medical resources. At the same time, a large number of medical workers have been infected after the epidemic was fully lifted, further straining medical resources. At this point, the government is unable to provide special care for the elderly, even though it knows that the elderly have weak resistance.

2. Home medical service mode. When the Omicrone strain of the disease hit China, it caused a run on hospitals. And the elderly, due to their poor health, do not have the energy to go to the hospital to participate in the queue. At this point, home medical services appear very necessary. Different from the medical model without special care, home medical services focus on the elderly in a region, providing home medical care for the elderly, changing the focus of epidemic prevention and control from preventing the elderly from getting infected to protecting their health and preventing them from becoming seriously ill. So as to protect the health of the elderly. For example, Xiangmihu Street in Futian District, Shenzhen, China, conducts accurate screening and focuses on the elderly and other vulnerable groups in the district. And the local government collects information about the physical condition of the elderly. The government establishes effective one-to-one contact with older people with underlying medical conditions. Diagnosis and treatment should be provided to the elderly in case of physical abnormalities caused by infection with Omicron.

3. "Green channel" medical service mode. Older people are more vulnerable to the pandemic. The elderly are more likely to suffer from severe or critical illness. In order to facilitate the treatment of the elderly, some hospitals provide "green channel" for the elderly group. Under this model, the local government establishes smooth communication channels with the elderly groups, so as to guarantee timely and effective treatment for the elderly infected in residential areas. The hospital timely understands the health status of the elderly nearby, and provides one-to-one health management and guidance for the elderly with basic classes. When the elderly are in serious condition, the critically ill elderly shall be transferred to the third-level hospital under the jurisdiction for follow-up treatment.

The relationship between the three medical service modes is shown in Fig 1.

**2.1.2 Hypothesis.** During the COVID-19 pandemic, a close relationship has been established between the government and the elderly community. The five Central Asian countries have taken measures to ensure the physical health of the elderly. The government and elderly representatives can communicate and consult on an equal basis to discuss and solve problems related to the health of the elderly. The following are some specific actions taken by the government in this process. First, provide information. The government provides information about COVID-19 to the elderly through various channels, such as television, the Internet, telephone, and community activities, including information about vaccination, methods to prevent infection, early warning signs of symptoms, etc. [26]. Second, provide medical resources. The government provides necessary resources for the elderly, such as masks, hand sanitizers, disinfectants, etc., and ensures that they can receive COVID-19 vaccines. In some regions, the government also provides telephone and online counseling services for the elderly and their caregivers. Third, develop policies. The government has developed some policies for the elderly to ensure their health [27]. For example, some governments give priority to the elderly to provide COVID-19 vaccines, or provide home vaccination services for the elderly who cannot go out. In general, the relationship between the government and the elderly population has

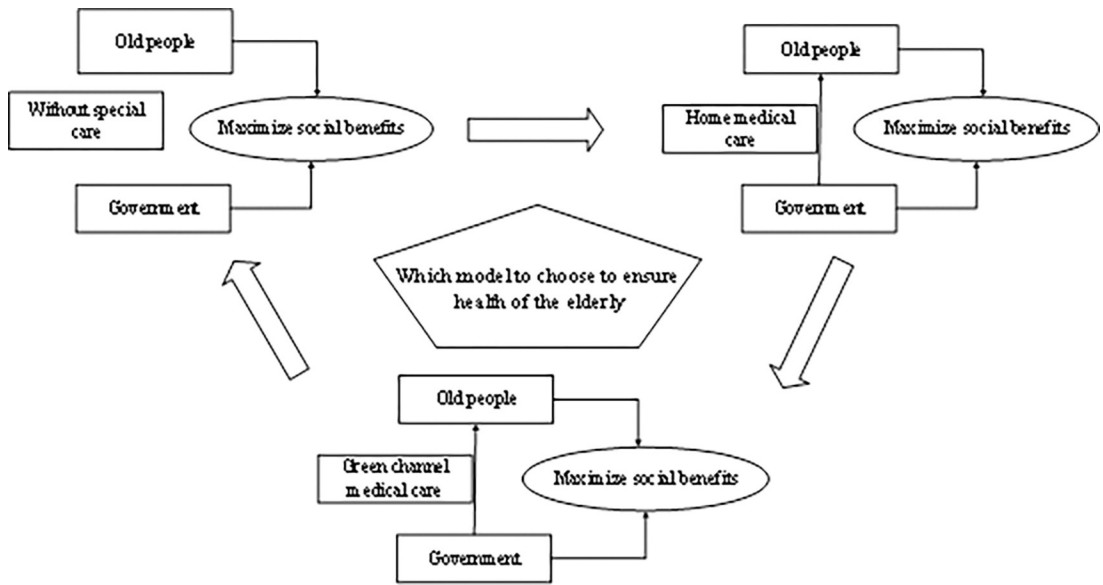

**Fig 1. Relationship between three different medical service modes.**

become closer during the COVID-19 pandemic, with the government taking a series of actions to protect the physical health of the elderly population.

In order to conduct in-depth research and comparative analysis of different medical service models in the five Central Asian countries, we need to find out their respective advantages, disadvantages and applicable conditions. In this way, we can adjust the allocation of medical resources to the elderly population, optimize medical policies, and help the treatment of the elderly population achieve more ideal results. In order to solve the above problems, the following hypothesis is established in this paper.

1. Only the government and the elderly are considered. There are not only old people in a country, but also many children and young people. The same applies to five Central Asian countries. Children and young people also have medical needs when COVID-19 is more severe. Overtreatment of the elderly can squeeze the young and children, resulting in dissatisfaction among children and young people. However, the virus has hurt older people more than children and young people, resulting in higher death rates. In order to better reduce the public health crisis caused by the novel coronavirus epidemic and maximize the protection of people's health, this article only considers the existence of the government and the elderly group of two players. The variables, state variables, and utility functions are all centered around these two players.

2. Medical personnel in five Central Asian countries are relatively sufficient. Five Central Asian countries, whose economies are developing more slowly, do not have enough money to upgrade medical equipment and buy specific drugs for COVID-19. Before the collapse of the Soviet Union, however, the Soviet population was relatively well educated and produced a large number of medical staff. Despite the collapse of the Soviet Union, five Central Asian countries have relatively abundant medical staff. In the face of COVID-19, five Central Asian countries have relatively backward medical resources such as equipment and medicines [15]. These countries can help by improving their models of care and by having more health workers. In this paper, one of the independent variables related to the elderly

population is assumed to be the amount of medical services provided by medical staff to the elderly.

3. Government decision-making, access to medical resources and the degree of infection of the elderly are in dynamic change. During the COVID-19 pandemic, the virus has been mutating. Every day, some elderly people contract the virus, while others recover. In this paper, another independent variable associated with the elderly population is assumed to be the degree of infection among the elderly. In order to protect the health of the elderly, the governments of five Central Asian countries will make relevant decisions according to the illness and death of the elderly in their own countries. Thus, the independent variable related to the government is assumed to be the input of medical resources. Government decisions further affect the availability of care for older people, which in turn affects morbidity and mortality. Over time, this cycle continues, leading to dynamic changes in government decision-making, access to health care resources for older persons and levels of infection.

4. Five Central Asian countries have accurate information about their elderly. At the beginning of the COVID-19 outbreak, countries around the world responded with all-out efforts. Five Central Asian countries are no exception. In order to prevent and control the epidemic, local health departments earnestly do a good job of local investigation work, more prepared to grasp the information of the elderly. The information includes where the elderly live, their underlying medical conditions, and the extent of their illness. Five Central Asian countries with this information, the government can carry out timely treatment of local elderly people.

**2.1.3 Variable definition.** These parameters and variables are defined as shown in Table 1.

## 2.2 Differential game of different medical service models

Under the COVID-19 pandemic, in order to protect the health of the elderly, the five Central Asian countries can adopt three modes: no special care, home medical care, and "green channel" medical care. In order to compare the scope of use of different modes, this paper constructs a differential game model. Among them, utility function and state variable are two important components of the differential game framework [28]. In the differential game, "utility function" represents the interests of the game participant (i.e., the player), that is, his objective function. The goal of each participant is usually to maximize his utility function. The utility function can depend on the action, state, and time of the participant. In different game scenarios, utility function can represent different things. For example, for a company, utility function may represent profit or market share; while for a consumer, utility function may represent consumption satisfaction or the number of products obtained. In the differential game, "state variable" describes the state of the game system at a certain point in time. The change of state variable is affected by the control variable selected by the participant (i.e., the strategy selected by the participant) [29]. For example, the company's market share can be regarded as a state variable, while the company's advertising investment, pricing strategy, etc. can be regarded as a control variable. According to the characteristics of differential game, the derivative of state variable (i.e. the rate of change over time) can be expressed as a function of one or more control variables, representing the change law of system state under a given strategy. In general, under the framework of differential game, each participant selects the optimal control variable (or strategy) to influence the change of state variable, so as to maximize its own utility function.

**Table 1. The main definition of variables and parameters in this article.**

| variables and parameters | specific meaning |
|---|---|
| $Y = \{N,H,G\}$ | three modes of medical service (no special care, home medical care, "green channel" medical care) |
| *independent variable* | |
| $M_{Y1}(t)$ | medical services received by the elderly under medical service model $Y$ |
| $I_{Y1}(t)$ | the extent to which the elderly are infected under medical service model $Y$ |
| $F_{Y2}(t)$ | the government's medical resource input under medical service model $Y$ |
| $x_{Y1}(t)$ | health status of the elderly under medical service model $Y$ |
| $x_{Y2}(t)$ | the credibility of the government under medical service model $Y$ |
| *parameter* | |
| $\rho$ | the discount rate occurring over time, $0 \leq \rho \leq 1$ |
| $\delta_1$ | the decay rate of satisfaction of the elderly, $\delta_1 > 0$ |
| $\delta_2$ | the decay rate of the government's credibility, $\delta_2 > 0$ |
| $b$ | the benefits of unit medical services for the elderly, $b > 0$ |
| $b_G$ | the benefits of providing "green channel" to the elderly, $b_G > 0$ |
| $l_1$ | the positive impact of unit satisfaction, $l_1 > 0$ |
| $l_2$ | the positive impact of the credibility of the unit government, $l_2 > 0$ |
| $c_I$ | damage to the body caused by infections in the elderly, $c_I > 0$ |
| $c_M$ | cost per unit of health care for older persons, $c_M > 0$ |
| $c_F$ | the cost of a unit of medical resources invested by the government, $c_F > 0$ |
| $c_H$ | additional health care costs associated with home medical care for the elderly, $c_H > 0$ |
| $c_G$ | the additional health costs of providing "green channel" to the elderly, $c_G > 0$ |
| $\alpha_1$ | reduced cross-infection among the elderly in the home medical care model, $\alpha_1 > 0$ |
| $\lambda_1$ | dissatisfaction with the infection of the elderly, $\lambda_1 > 0$ |
| $\lambda_2$ | the credibility gained by the government for investing unit medical resources, $\lambda_2 > 0$ |
| $\lambda_3$ | the increase in health care costs without special care because the number of infections increases, $\lambda_3 > 0$ |
| $\lambda_4$ | the percentage who didn't get treatment because they were in line, $\lambda_4 > 0$ |
| $\lambda_H$ | the extra credibility of government for providing home health care, $\lambda_H > 0$ |
| $\lambda_G$ | the extra credibility of government for providing "green channel" health care, $\lambda_G > 0$ |
| $\beta$ | the improvement in physical health after elderly get from receiving treatment, $\beta > 0$ |
| $\beta_H$ | elderly people get more improvement in physical health from home medical cares, $\beta_H > 0$ |
| $\beta_G$ | the elderly get more improvement in physical health from enjoying the "green channel" of medical treatment, $\beta_G > 0$ |
| *function* | |
| $J_{Y1}(t)$ | social welfare function of the elderly under the medical service model $Y$ |
| $J_{Y2}(t)$ | the government's social welfare function under the medical service model $Y$ |
| $V_{Y1}(t)$ | social benefits of the elderly under the medical service model $Y$ |
| $V_{Y2}(t)$ | social benefits of the government under the medical service model $Y$ |

**2.2.1 No special care.** In the no special care mode, the social welfare functions of the elderly group and the government are:

$$J_{N1} = \int_0^\infty [bM_{N1}(t) - c_I I_{N1}(t) - \frac{c_M}{2}M_{N1}^2(t) + l_1 x_{N1}(t)]e^{-\rho t}dt \tag{1}$$

$$J_{N2} = \int_0^\infty \left[ -\frac{c_F}{2}F_{N2}^2(t) + l_2 x_{N2}(t) \right]e^{-\rho t}dt \tag{2}$$

In the above formula, $bM_{N1}(t)$ represents the income of the elderly due to receiving medical services without special care. $c_I I_{N1}(t)$ represents the damage to the body caused by virus infection in the elderly without special care mode. $\frac{c_M}{2}M_{N1}^2(t)$ represents the medical costs of treating the elderly. $l_1 x_{N1}(t)$ indicates the psychological satisfaction of the elderly due to treatment. $\frac{c_F}{2}F_{N2}^2(t)$ represents the cost of medical resources invested by the government. $l_2 x_{N2}(t)$ represents the government's credibility for fighting the pandemic.

In the no special care mode, the changes of health status of the elderly can be expressed as:

$$\dot{x}_{N1}(t) = -\lambda_1 I_{N1}^2(t) + \ln(\beta + 1)M_{N1}(t) - \delta x_{N1}(t) \tag{3}$$

In the no special care mode, the changes of credibility of government can be expressed as:

$$\dot{x}_{N2}(t) = (\lambda_2 + \lambda_3 - \lambda_4)F_{N2}(t) - \delta x_{N2}(t) \tag{4}$$

In the above formula, $\lambda_1 I_{N1}^2(t)$ represents the bad effects on the health caused by the infection of the elderly. $\ln(\beta+1)M_{N1}(t)$ represents the old man's health improved after treatment. $\delta x_{N1}(t)$ represents the attenuation of the satisfaction degree of the elderly. $(\lambda_2+\lambda_3-\lambda_4)F_{N2}(t)$ represents the credibility of the government for investing in health care. $\delta x_{N2}(t)$ represents the decline of the government's credibility.

**2.2.2 Home medical service.** In the home medical service mode, the social welfare functions of the elderly group and the government are:

$$J_{H1} = \int_0^\infty \left[ bM_{H1}(t) - (c_I - \alpha_1)I_{H1}(t) - \frac{c_M + c_H}{2}M_{H1}^2(t) + l_1 x_{H1}(t) \right] e^{-\rho t} dt \tag{5}$$

$$J_{H2} = \int_0^\infty \left[ -\frac{c_F + c_H}{2}F_{H2}^2(t) + l_2 x_{H2}(t) \right] e^{-\rho t} dt \tag{6}$$

In the above formula, $bM_{H1}(t)$ represents the income that the elderly get from receiving medical services under the home-based medical service model. $(c_I - \alpha_I)I_{H1}(t)$ represents the damage to the body caused by virus infection in the elderly under the model of home medical care. $\frac{c_M + c_H}{2}M_{H1}^2(t)$ represents the medical costs of treating the elderly. $l_1 x_{H1}(t)$ indicates the psychological satisfaction of the elderly due to treatment. $\frac{c_F + c_H}{2}F_{H2}^2(t)$ represents the cost of medical resources invested by the government. $l_2 x_{H2}(t)$ represents the credibility the government gained for fighting the pandemic.

In the home medical service mode, the changes of health status of the elderly can be expressed as:

$$\dot{x}_{H1}(t) = -\lambda_1 I_{H1}^2(t) + \ln(\beta + \beta_H + 1)M_{H1}(t) - \delta x_{H1}(t) \tag{7}$$

In the home medical service mode, the changes of credibility of government can be expressed as:

$$\dot{x}_{H2}(t) = (\lambda_2 + \lambda_H)F_{H2}(t) - \delta x_{H2}(t) \tag{8}$$

In the above formula, $\lambda_1 I_{H1}^2(t)$ means the bad effects on the health caused by the infection of the elderly. $\ln(\beta+\beta_H+1)M_{H1}(t)$ represents the old man's health improved after treatment. $\delta x_{H1}(t)$ represents a decline in the satisfaction of the elderly. $(\lambda_2+\lambda_H)F_{H2}(t)$ represents the credibility gained by the government for investing medical resources under the home health care model. $\delta x_{N2}(t)$ represents a decline in the government's credibility.

**2.2.3 "Green channel" medical service.** In the "green channel" medical service mode, the social welfare functions of the elderly group and the government are:

$$J_{G1} = \int_0^\infty \left[ (b + b_G)M_{G1}(t) - c_I I_{G1}(t) - \frac{c_M + c_G}{2} M_{G1}^2(t) + l_1 x_{G1}(t) \right] e^{-\rho t} dt \tag{9}$$

$$J_{G2} = \int_0^\infty \left[ -\frac{c_F + c_G}{2} F_{G2}^2(t) + l_2 x_{G2}(t) \right] e^{-\rho t} dt \tag{10}$$

In the above formula, $(b+b_G)M_{G1}(t)$ represents the income of the elderly due to receiving medical services under the "green channel" mode. $c_I I_{G1}(t)$ represents the damage to the body caused by viral infections in the elderly in the "green channel" mode. $\frac{c_M + c_G}{2} M_{G1}^2(t)$ represents the medical costs of treating the elderly. $l_1 x_{G1}(t)$ indicates the psychological satisfaction of the elderly as a result of treatment. $\frac{c_F + c_G}{2} F_{G2}^2(t)$ represents the cost of health care resources invested by the government. $l_2 x_{G2}(t)$ represents the government's credibility for fighting the epidemic.

In the "green channel" medical service mode, the changes of health status of the elderly can be expressed as:

$$\dot{x}_{G1}(t) = -\lambda_1 I_{G1}^2(t) + \ln(\beta + \beta_G + 1)M_{G1}(t) - \delta x_{G1}(t) \tag{11}$$

In the "green channel" medical service mode, the changes of credibility of government can be expressed as:

$$\dot{x}_{G2}(t) = (\lambda_2 + \lambda_G)F_{G2}(t) - \delta x_{G2}(t) \tag{12}$$

In the above formula, $\lambda_1 I_{G1}^2(t)$ means the bad effects on the health caused by the infection of the elderly. $\ln(\beta + \beta_G + 1)M_{G1}(t)$ represents the old man's health improved after treatment in the "green channel" mode. $\delta x_{G1}(t)$ represents the attenuation of the satisfaction degree of the elderly. $(\lambda_2 + \lambda_G)F_{G2}(t)$ represents the credibility gained by the government's investment of medical resources under the "green channel" mode. $\delta x_{G2}(t)$ represents the decline of the government's credibility.

## 3. Results

In the differential game, the social benefits of the elderly and the government in five Central Asian countries are not only affected by control variables and parameters, but also constantly change with time, state and state's impact on social welfare. In order to better calculate the medical resources obtained by the elderly in five Central Asian countries, the infection degree of the elderly, the government's investment in medical resources and social benefits, the HJB formula is adopted. HJB formula is a partial differential equation, which is the core of optimal control.

### 3.1 HJB formula

If the elderly group does not receive any special care, then the HJB equation of the social welfare function of the elderly and the government in this mode is:

$$\rho V_{N1} = \max_{M_{N1}(t), I_{N1}(t)} \left\{ \left[ bM_{N1}(t) - c_I I_{N1}(t) - \frac{c_M}{2} M_{N1}^2(t) + l_1 x_{N1}(t) \right] \right.$$
$$\left. + \frac{\partial V_{N1}}{\partial x_{N1}} \left[ -\lambda_1 I_{N1}^2(t) + \ln(\beta + 1)M_{N1}(t) - \delta x_{N1}(t) \right] \right\} \tag{13}$$

$$\rho V_{N2} = \max_{F_{N2}(t)} \left\{ \left[ -\frac{c_F}{2} F_{N2}^2(t) + l_2 x_{N2}(t) \right] + \frac{\partial V_{N2}}{\partial x_{N2}} [(\lambda_2 + \lambda_3 - \lambda_4) F_{N2}(t) - \delta x_{N2}(t)] \right\} \quad (14)$$

If the elderly in five Central Asian countries receive home medical services, then the HJB equation of the social welfare function of the elderly and the government in this mode is:

$$\rho V_{H1} = \max_{M_{H1}(t), I_{H1}(t)} \left\{ \left[ b M_{H1}(t) - (c_I - \alpha_1) I_{H1}(t) - \frac{c_M + c_H}{2} M_{H1}^2(t) + l_1 x_{H1}(t) \right] \right.$$
$$\left. + \frac{\partial V_{H1}}{\partial x_{H1}} \left[ -\lambda_1 I_{H1}^2(t) + \ln(\beta + \beta_H + 1) M_{H1}(t) - \delta x_{H1}(t) \right] \right\} \quad (15)$$

$$\rho V_{H2} = \max_{F_{H2}(t)} \left\{ \left[ -\frac{c_F + c_H}{2} F_{H2}^2(t) + l_2 x_{H2}(t) \right] + \frac{\partial V_{H2}}{\partial x_{H2}} [(\lambda_2 + \lambda_H) F_{H2}(t) - \delta x_{H2}(t)] \right\} \quad (16)$$

If hospitals in five Central Asian countries provide "green channel" for the elderly, then the HJB equation of the social welfare function of the elderly and the government in this mode is:

$$\rho V_{G1} = \max_{M_{G1}(t), I_{G1}(t)} \left\{ \left[ (b + b_G) M_{G1}(t) - c_I I_{G1}(t) - \frac{c_M + c_G}{2} M_{G1}^2(t) + l_1 x_{G1}(t) \right] \right.$$
$$\left. + \frac{\partial V_{G1}}{\partial x_{G1}} \left[ -\lambda_1 I_{G1}^2(t) + \ln(\beta + \beta_G + 1) M_{G1}(t) - \delta x_{G1}(t) \right] \right\} \quad (17)$$

$$\rho V_{G2} = \max_{F_{G2}(t)} \left\{ \left[ -\frac{c_F + c_G}{2} F_{G2}^2(t) + l_2 x_{G2}(t) \right] + \frac{\partial V_{G2}}{\partial x_{G2}} [(\lambda_2 + \lambda_G) F_{G2}(t) - \delta x_{G2}(t)] \right\} \quad (18)$$

### 3.2 Result of equilibrium

Proposition 1: In the model without special care, medical resources received by the elderly in five Central Asian countries, infections suffered by the elderly, medical resources invested by the government, social benefits of the elderly and social benefits of the government are respectively:

$$M_{N1}^*(t) = \frac{1}{c_M} \left[ b + \frac{l_1}{\rho + \delta} \ln(\beta + 1) \right], \ I_{N1}^*(t) = -\frac{c_I}{2\lambda_1} \left( \frac{l_1}{\rho + \delta} \right)^{-1} \quad (19)$$

$$F_{N2}^*(t) = \frac{\lambda_2 + \lambda_3 - \lambda_4}{c_F} \frac{l_2}{\rho + \delta} \quad (20)$$

$$V_{N1}^* = \frac{l_1}{\rho + \delta} x_{N1} + \frac{1}{\rho} \left[ b \frac{1}{c_M} \left( b + \frac{l_1}{\rho + \delta} \ln(\beta + 1) \right) + c_I \frac{c_I}{2\lambda_1} \left( \frac{l_1}{\rho + \delta} \right)^{-1} - \right.$$

$$\frac{c_M}{2} \left( \frac{1}{c_M} \right)^2 \left( b + \frac{l_1}{\rho + \delta} \ln(\beta + 1) \right)^2 \Big]$$

$$+ \frac{l_1}{\rho + \delta} \frac{1}{\rho} \left[ -\lambda_1 \left( \frac{c_I}{2\lambda_1} \right)^2 \left( \frac{l_1}{\rho + \delta} \right)^{-2} + \ln(\beta + 1) \frac{1}{c_M} \left( b + \frac{l_1}{\rho + \delta} \ln(\beta + 1) \right) \right] \tag{21}$$

$$V_{N2}^* = \frac{l_2}{\rho + \delta} x_{N2} - \frac{1}{\rho} \frac{c_F}{2} \left( \frac{\lambda_2 + \lambda_3 - \lambda_4}{c_F} \right)^2 \left( \frac{l_2}{\rho + \delta} \right)^2 + \frac{l_2}{\rho + \delta} \frac{1}{\rho} (\lambda_2 + \lambda_3 - \lambda_4)^2 \frac{1}{c_F} \left( \frac{l_2}{\rho + \delta} \right) \tag{22}$$

Conclusion 1: Under the mode of no special care, the medical services obtained by the elderly in five Central Asian countries are in direct proportion to the benefits brought by medical services, inversely proportional to the cost of medical services and proportional to the satisfaction obtained from treatment. The extent to which the elderly are infected is inversely proportional to the physical damage caused by the infection and proportional to the dissatisfaction caused by the infection. The government's input of medical resources is directly proportional to the credibility brought by the input of medical resources, and inversely proportional to the cost of medical resources.

Proposition 2: In the home medical service mode, medical resources received by the elderly in five Central Asian countries, infections suffered by the elderly, medical resources invested by the government, social benefits of the elderly and social benefits of the government are respectively:

$$M_{H1}^*(t) = \frac{1}{c_M + c_H} \left[ b + \frac{l_1}{\rho + \delta} \ln(\beta + \beta_H + 1) \right], \quad I_{H1}^*(t) = -\frac{c_I - \alpha_1}{2\lambda_1} \left( \frac{l_1}{\rho + \delta} \right)^{-1} \tag{23}$$

$$F_{H2}^*(t) = \frac{\lambda_2 + \lambda_H}{c_F + c_H} \frac{l_2}{\rho + \delta} \tag{24}$$

$$V_{H1}^* = \frac{l_1}{\rho + \delta} x_{H1} + \frac{1}{\rho} \left[ \frac{(c_I - \alpha_1)^2}{2\lambda_1} \left( \frac{l_1}{\rho + \delta} \right)^{-1} - \frac{1}{2} \frac{1}{c_M + c_H} \left[ b + \frac{l_1}{\rho + \delta} \ln(\beta + \beta_H + 1) \right]^2 \right]$$

$$+ \frac{1}{\rho} b \frac{1}{c_M + c_H} \left[ b + \frac{l_1}{\rho + \delta} \ln(\beta + \beta_H + 1) \right] - \frac{l_1}{\rho + \delta} \frac{1}{\rho} \lambda_1 \left( \frac{c_I - \alpha_1}{2\lambda_1} \right)^2 \left( \frac{l_1}{\rho + \delta} \right)^{-2} \tag{25}$$

$$+ \frac{l_1}{\rho + \delta} \frac{1}{\rho} \ln(\beta + \beta_H + 1) \frac{1}{c_M + c_H} \left[ b + \frac{l_1}{\rho + \delta} \ln(\beta + \beta_H + 1) \right]$$

$$V_{H2}^* = \frac{l_2}{\rho + \delta} x_{H2} - \frac{1}{\rho} \frac{c_F + c_H}{2} \left( \frac{\lambda_2 + \lambda_H}{c_F + c_H} \frac{l_2}{\rho + \delta} \right)^2 + \frac{l_2}{\rho + \delta} \frac{1}{\rho} \frac{(\lambda_2 + \lambda_H)^2}{c_F + c_H} \frac{l_2}{\rho + \delta} \tag{26}$$

Conclusion 2: Under the mode of the elderly receiving home medical services, the medical services obtained by the elderly in five Central Asian countries are inversely proportional to the medical cost generated by the more home services, and directly proportional to the satisfaction generated by the more home services. Home medical care for older people can reduce cross-infection among older people. The government's input of medical resources is directly

proportional to the credibility generated by providing home service, and inversely proportional to the cost generated by providing home service.

Proposition 3: In the "green channel" medical service mode, medical resources received by the elderly in five Central Asian countries, infections suffered by the elderly, medical resources invested by the government, social benefits of the elderly and social benefits of the government are respectively:

$$M_{G1}^*(t) = \frac{1}{c_M + c_G}\left[b + b_G + \frac{l_1}{\rho + \delta}\ln(\beta + \beta_G + 1)\right], \ I_{G1}^*(t) = -\frac{c_I}{2\lambda_1}\left(\frac{l_1}{\rho + \delta}\right)^{-1} \quad (27)$$

$$F_{G2}^*(t) = \frac{\lambda_2 + \lambda_G}{c_F + c_G}\frac{l_2}{\rho + \delta} \quad (28)$$

$$V_{G1}^* = \frac{l_1}{\rho + \delta}x_{G1} + \frac{1}{\rho}\left[(b + b_G)\frac{1}{c_M + c_G}\left[b + b_G + \frac{l_1}{\rho + \delta}\ln(\beta + \beta_G + 1)\right]\right.$$

$$+ c_I\frac{c_I}{2\lambda_1}\left(\frac{l_1}{\rho + \delta}\right)^{-1} - \frac{1}{2}\frac{1}{c_M + c_G}\left[b + b_G + \frac{l_1}{\rho + \delta}\ln(\beta + \beta_G + 1)\right]^2\right]$$

$$+ \frac{l_1}{\rho + \delta}\frac{1}{\rho}\qquad \left[-\lambda_1\left(\frac{c_I}{2\lambda_1}\right)^2\left(\frac{l_1}{\rho + \delta}\right)^{-2}\right.$$

$$+ \ln(\beta + \beta_G + 1)\frac{1}{c_M + c_G}\left[b + b_G + \frac{l_1}{\rho + \delta}\ln(\beta + \beta_G + 1)\right]\right] \quad (29)$$

$$V_{G2}^* = \frac{l_2}{\rho + \delta}x_{G2} - \frac{1}{\rho}\frac{c_F + c_G}{2}\left(\frac{\lambda_2 + \lambda_G}{c_F + c_G}\frac{l_2}{\rho + \delta}\right)^2 + \frac{l_2}{\rho + \delta}\frac{1}{\rho}(\lambda_2 + \lambda_G)\frac{\lambda_2 + \lambda_G}{c_F + c_G}\frac{l_2}{\rho + \delta} \quad (30)$$

Conclusion 3: Under the mode of providing "green channel" medical services for the elderly, the medical services obtained by the elderly in five Central Asian countries are inversely proportional to the medical cost generated by more "green channel", and directly proportional to the satisfaction generated by more "green channel". Providing "green channel" services to the elderly does not reduce cross infection among the elderly. The input of medical resources by the government is directly proportional to the credibility and inversely proportional to the cost of providing "green channel".

### 3.3 Numerical analysis

This article assumes that the discount factor $\rho$ is 0.9. The decay rate $\delta_1$ of the satisfaction degree of the elderly and the decay rate $\delta_2$ of the credibility of the government are both 0.1. The benefit $b$ brought by unit medical service to the elderly is 3. The positive impact $l_1$ brought by unit satisfaction is 1. And the positive impact $l_2$ brought by unit government credibility is 1. The $c_I$ of damage to the body caused by infection in the elderly is 1.5. The cost $c_M$ of unit medical service for the elderly is 2. The cost $c_F$ of government input per unit of medical resources is 2. The improvement $\beta$ in physical health after elderly get from receiving treatment is 3.49. Both health status $x_1$ of the elderly and the credibility $x_2$ of the government are 1. The dissatisfaction $\lambda_1$ caused by the infection of the elderly is 1. The credibility $\lambda_2$ of the government for the unit of medical resources invested is 1. The increase in health care costs $\lambda_3$ without special care because the number of infections increases is 2. The percentage $\lambda_4$ who didn't get treatment

because they were in line is 2. The elderly people get more improvement $\beta_H$ in physical health from home medical cares is 28.7. The elderly get more improvement $\beta_G$ in physical health from enjoying the "green channel" of medical treatment is 7.71. Therefore, this article can calculate:

$$V_{H1}^* = 6.63 \tag{31}$$

When the reduction of cross infection $\alpha_1$ in the elderly under home medical care mode is 0.5,

$$V_{H1}^* = 1.28 + \frac{1}{2 + c_H} \times 23.44 \tag{32}$$

When the reduction of cross infection $\alpha_1$ in the elderly under home medical care mode is 1,

$$V_{H1}^* = 1.07 + \frac{1}{2 + c_H} \times 23.44 \tag{33}$$

The following graph (named Fig 2) can also be produced:

Conclusion 4: When the additional medical cost of home medical care for the elderly is small, the social benefits of the elderly under the home medical care mode are greater than those without special care mode. While the additional medical costs associated with home medical care increase gradually, the social benefits to the elderly decrease gradually. And in the end it's less than the social benefit of the no-special care model. At the same time, the greater the reduction of cross-infection in the elderly under the home medical service mode, the smaller the social benefits of the elderly.

When the extra income $b_G$ from providing "green channel" for the elderly is 1,

$$V_{G1}^* = 1 + \frac{1}{2 + c_G} \times 23.47 \tag{34}$$

When the extra income $b_G$ from providing "green channel" for the elderly is 0.5,

$$V_{G1}^* = 1 + \frac{20}{2 + c_G} \tag{35}$$

The following graph (named Fig 3) can also be produced:

Conclusion 5: When the extra medical cost of providing "green channel" for the elderly is relatively small, the social benefits of the elderly under the "green channel" mode are greater than those without special care mode. When the extra medical costs caused by the green medical treatment channel gradually increase, the social benefits of the elderly gradually decrease. And, in the end, it's less than the social benefit of the no-special care model.

$$V_{N2}^* = 1.278 \tag{36}$$

When the government gets a good credibility $\lambda_H$ for providing home medical care is 0.3,

$$V_{H2}^* = 1 + 0.94 \times \frac{1}{2 + c_H} \tag{37}$$

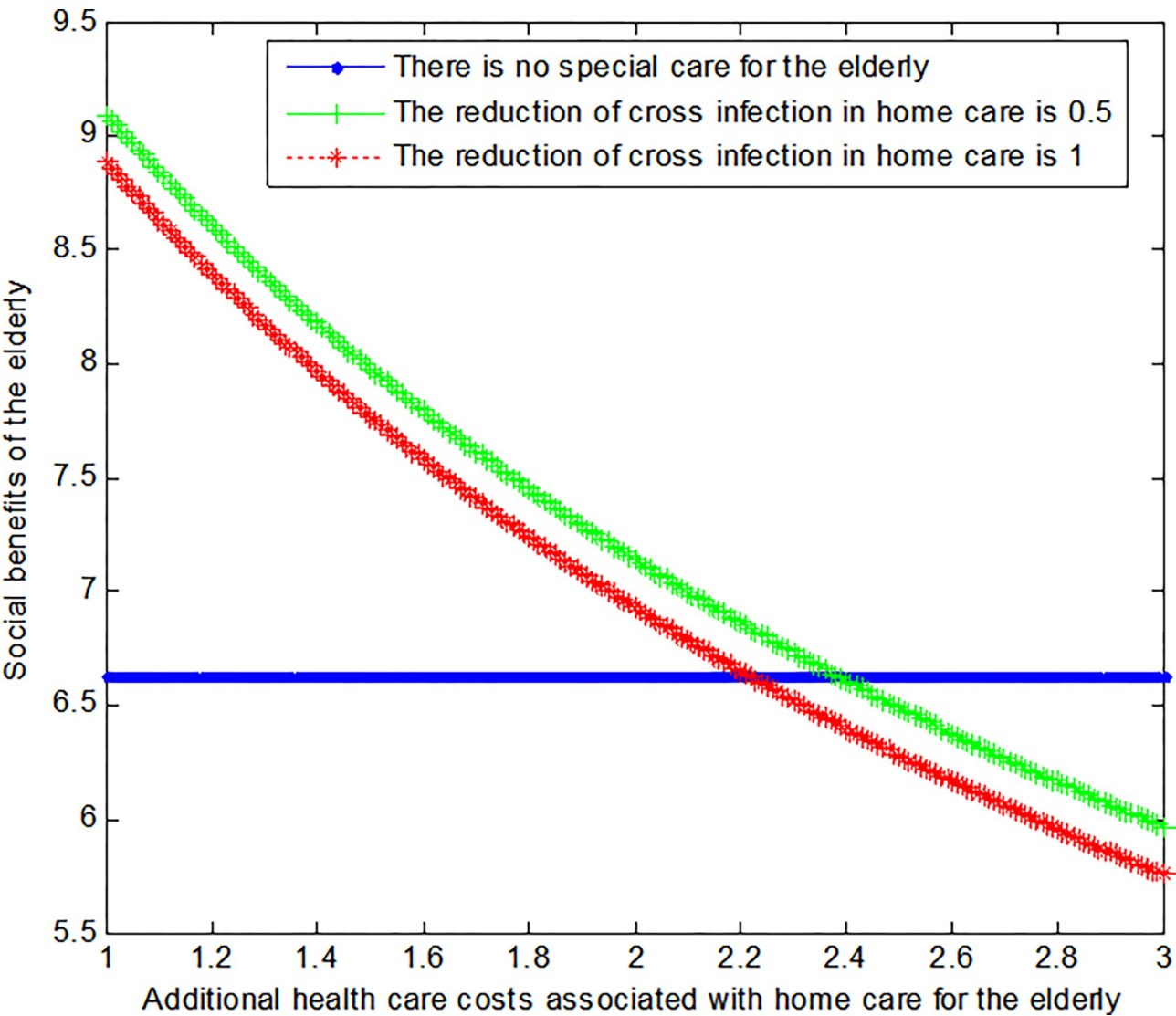

**Fig 2. Impact of additional health care costs on elderly' social welfare.**

When the government gets a good credibility $\lambda_H$ for providing home medical care is 0.5,

$$V_{H2}^* = 1 + 1.25 \times \frac{1}{2 + c_H} \tag{38}$$

The following graph (named Fig 4) can also be produced:

Conclusion 6: When the extra medical cost of home medical care for the elderly is relatively small, the social benefits obtained by the government under the home medical care mode are greater than those without special care mode. When the extra medical costs caused by home medical care gradually increase, the social benefits obtained by the government gradually decrease, and finally it is less than the social benefits without special care mode. At the same time, the greater the credibility of the government under the home medical service model, the smaller the social benefits of the government.

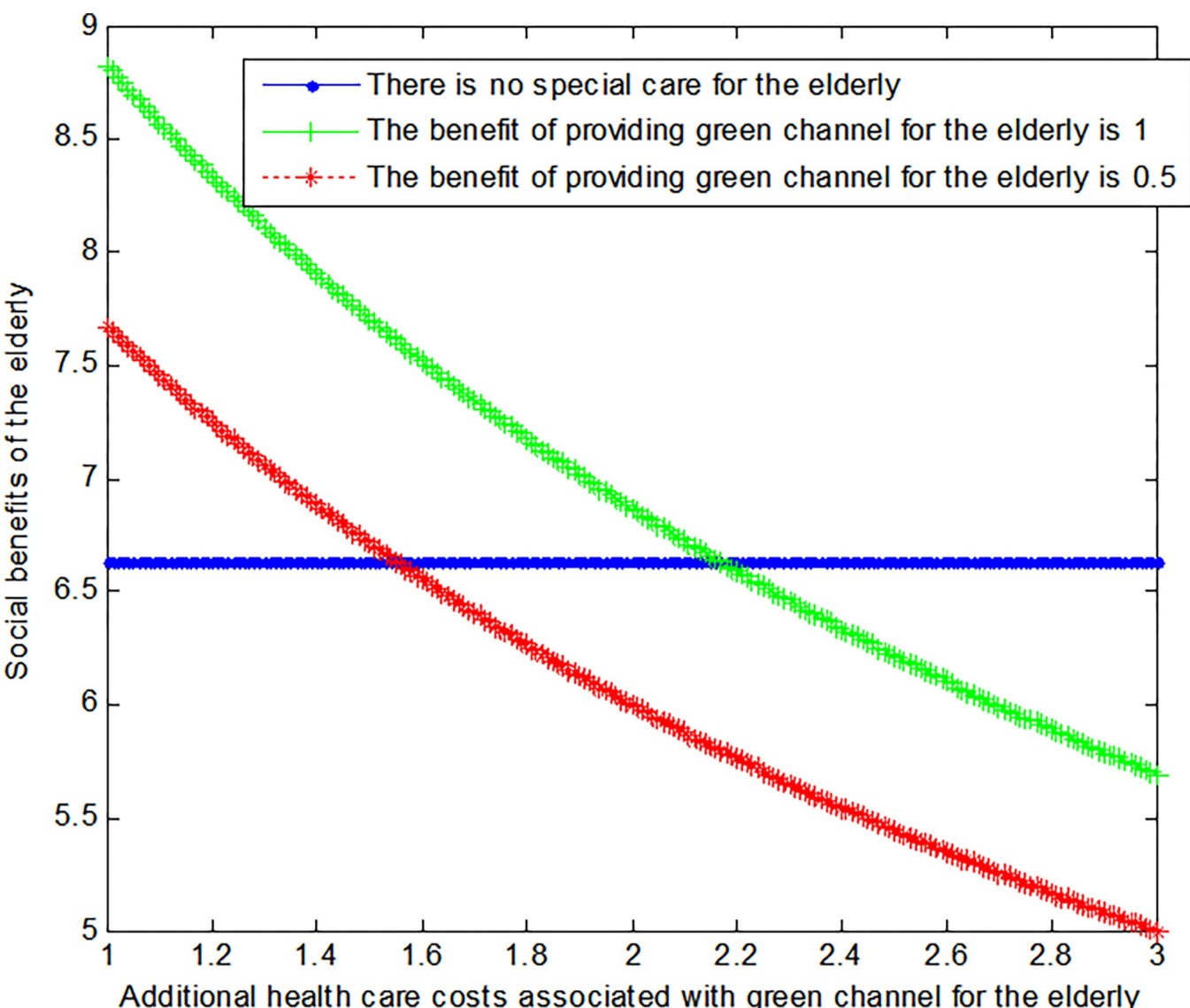

**Fig 3. Impact of additional health care costs on elderly' social welfare.**

When the government gets a good credibility $\lambda_G$ for providing "green channel" is 0.3,

$$V_{G2}^* = 1 + 0.94 \times \frac{1}{2 + c_G} \tag{39}$$

When the government gets a good credibility $\lambda_G$ for providing "green channel" is 0.4,

$$V_{G2}^* = 1 + 1.09 \times \frac{1}{2 + c_G} \tag{40}$$

The following graph (named Fig 5) can also be produced:

Conclusion 7: When providing the "green channel" for the elderly results in less medical costs, the social benefits obtained by the government under the "green channel" mode are greater than those without special care mode. When the extra medical costs caused by the

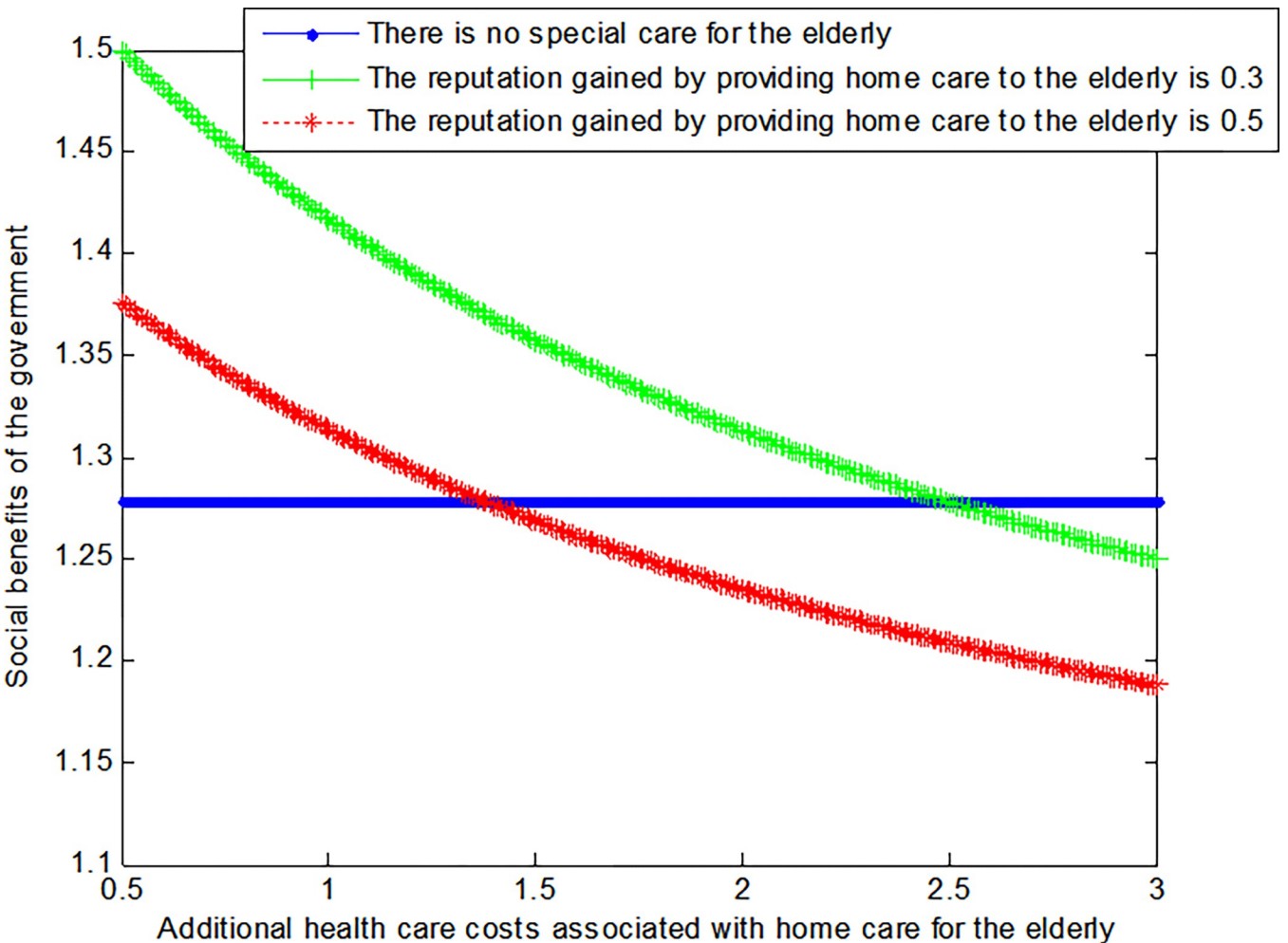

**Fig 4. Impact of additional health care costs on government' social welfare.**

"green channel" gradually increase, the social benefits obtained by the government gradually decrease, and finally it is less than the social benefits under the mode without special care. At the same time, the greater the credibility of the government under the "green channel" service mode, the greater the social benefits of the government.

## 4. Discussion

The novel coronavirus spreads quickly, and five Central Asian countries have relatively poor medical conditions. Five Central Asian countries need to improve their medical models to keep the elderly healthy. In order to maximize the health of the elderly and reduce the morbidity and mortality caused by the novel coronavirus, the governments of five Central Asian countries should provide home medical services for the elderly or provide green medical channels for the elderly. However, the cost of providing medical services to the elderly is higher, although it can avoid reinfection or cross-infection among the elderly. However, providing "green channel" for the elderly in hospitals is cheaper and allows them to receive better medical treatment than at home. But this makes it easy for the elderly to become reinfected with the virus. Therefore, the application scope of various medical service modes is an important issue

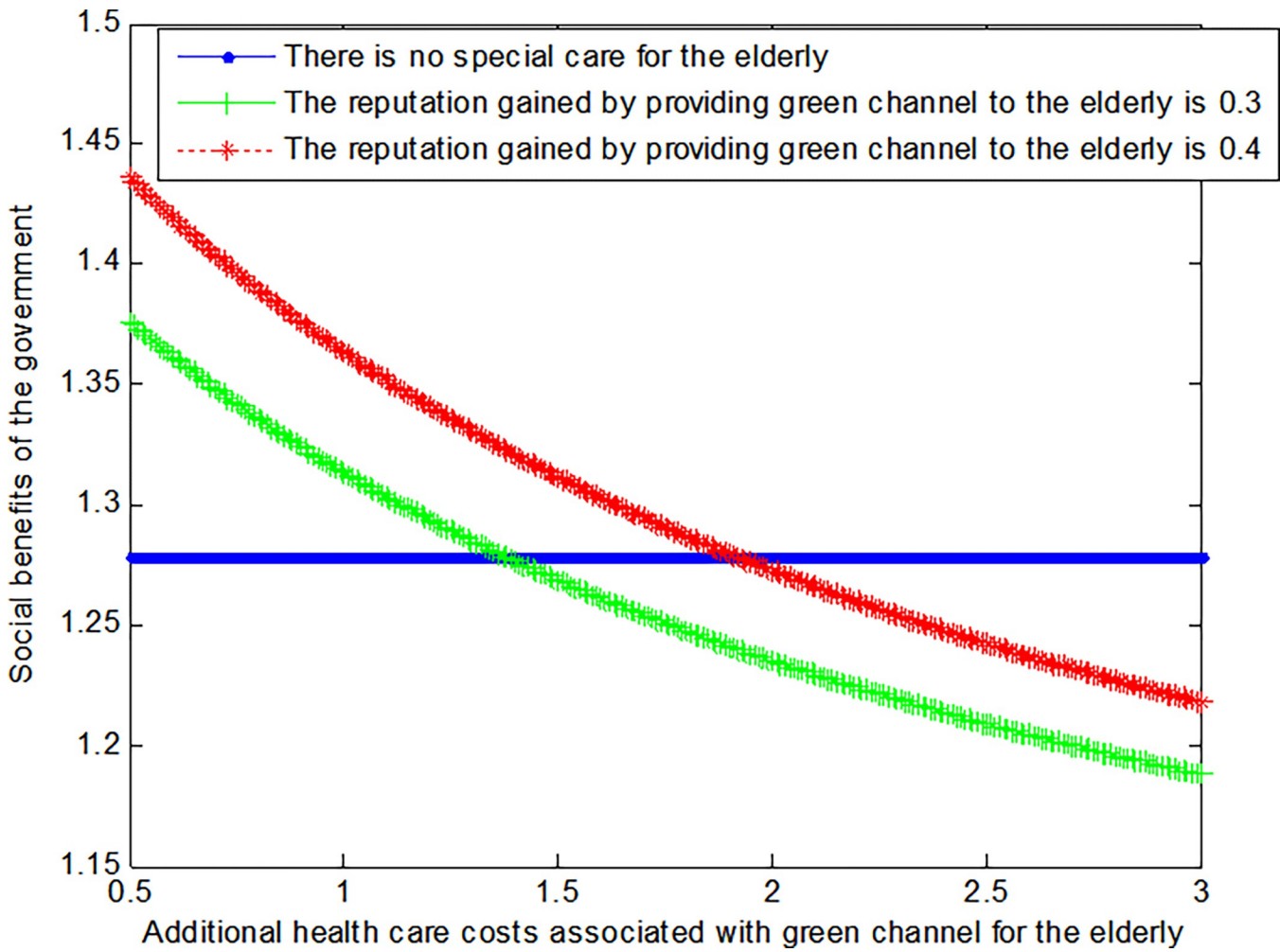

**Fig 5. Impact of additional health care costs on government' social welfare.**

in this article. The situation of the elderly, the course of the epidemic and the decision-making of governments are dynamic. For this reason, differential game is used in the field of anti-terrorism. In particular, considering that the government fully grasps the information of the elderly, how the government of five Central Asian countries can use the limited medical resources to ensure the health of the elderly to the maximum extent.

Due to the limitation of their own economic development, five Central Asian countries have relatively limited medical resources [30]. At this time, the governments of five Central Asian countries should insist on effective input and management, and completely change the unfair distribution of limited medical resources among different citizens and different regions. Under the COVID-19 pandemic, the elderly need more medical resources. Only in this way can the elderly survive the epidemic safely and increase their satisfaction with society. Due to the weak body resistance of the elderly, in order to reduce the incidence of the elderly, five Central Asian countries should also properly restrict the travel of the elderly.

Five Central Asian countries provide the elderly with "green channel" to medical services in their own hospitals, which can make medical treatment efficient and convenient for the elderly. When COVID-19 hit five Central Asian countries, hospitals were overwhelmed. At this time, five Central Asian countries should optimize the process of rapid pre-examination

for the elderly and shorten the waiting time outside the diagnosis area. And improve the elderly to provide access to medical efficiency, and help the elderly smooth passage. Although providing "green channel" to medical services for the elderly cannot reduce cross infection, it can enable the elderly to get timely and professional treatment when they fall ill. Thus, the elderly in five Central Asian countries are more satisfied with the society.

Older people have a higher proportion of underlying diseases than younger people. The novel coronavirus is likely to do great harm to the elderly with underlying diseases [31]. Can lead to serious complications in this population. When the elderly suffer from very serious complications, the extra medical costs of house calls or "green channel" are greater. At this point, neither home medical services nor "green channel" to medical services for the elderly will improve their benefits. At this point, five Central Asian countries would do better to direct their limited resources to the sicker elderly.

The greater the credibility of the government under the "green channel" service model, the greater the social benefits of the government in five Central Asian countries. However, the greater the credibility of the government under the home medical service model, the smaller the social benefits of the government in five Central Asian countries. This is mainly because the medical resources consumed by home medical services are greater than those consumed by "green channel". The greater the credibility of the government under the home medical service model, the more serious the development of the epidemic, many patients in need of treatment. Five Central Asian countries have a relatively concentrated population, with great differences between urban and rural areas [32]. At this time, if the elderly are provided with more home medical services, the government will gain less income. Therefore, when the epidemic is serious, the "green channel" mode should be adopted first.

The research of this paper can provide reference for the reform of the American government's healthcare system, which can be carried out in the following aspects. First, service quality and efficiency. By comparing different healthcare models, this paper finds out how to optimize service quality and efficiency. For the American healthcare system, by learning from the successful experience of other countries, it can balance the relationship between medical efficiency and medical quality, and improve efficiency while ensuring quality [33]. Second, macro-control and incentive mechanism. Different healthcare models have different macro-control means and incentive mechanisms. The United States can learn from the relevant experience of other countries, build a more sound incentive mechanism, reasonably adjust the price of medical services, and guide the supply and demand of medical services through macro-control to ensure the effective use of medical resources. Third, the role of public and private sectors. In different healthcare models, the public and private sectors play different roles. The United States can find the most suitable public-private healthcare service system by studying the models of other countries. In general, by translating the comparative analysis theory of different healthcare models into specific reform policies, the American government's healthcare system reform can achieve greater success. Meanwhile, the study can also provide some specific reference for the management of generic drugs in India. For example, the comparative analysis can reveal which treatment methods are more effective in the treatment of COVID-19. India can refer to these conclusions and select appropriate drug preparations for imitation, so as to treat COVID-19 more targeted. By analyzing the effects of different treatment modalities, India can be more clear about which drugs to imitate, or which pathological links should be focused on in scientific research, providing direction for the development of new drugs and drug imitation. Those treatment methods and drugs that have been proven effective in the comparative analysis can guide the rational investment of scientific research force in India.

## 5. Conclusion

Considering the rapid spread of the novel coronavirus, its great damage to the elderly and the limited medical resources in five Central Asian countries, this article constructs a differential game model with no special care, home medical services and "green channel". The equilibrium results are compared and analyzed. When the extra medical costs caused by home medical care or "green channel" gradually increase, the social benefits of the elderly and the government will gradually decrease, and eventually it is less than the social benefits of the model without special care. The greater the credibility of the government under the "green channel" service model, the greater the social benefits of the government. However, the greater the credibility of the government under the home medical service model, the smaller the social benefits of the government.

The research of this article can also be extended. For example, this article only considers the existence of two game parties: the elderly and the government. This article divides the special care for the elderly into two modes: home service and "green channel". This article assumes that the COVID-19 pandemic will take a larger toll on the health of the elderly. This article assumes that the government can fully grasp the information of the elderly. In future studies, it is possible to consider the existence of young people as a game, the existence of mixed care models, the damage of COVID-19 to everyone is not very different, and the government only has partial information about the elderly. And to carry on the relevant problem research. In addition, the study of this article is not only applicable to the health of the elderly in five Central Asian countries, but also has certain reference significance for how to effectively reform the medical system of the US government and how to regulate generic drugs in India. For example, this article can provide references for the reform of the US medical system in terms of service quality, macro-control, and the role of the public and private sectors. At the same time, India can draw on these conclusions to select suitable drug formulations for imitation, so as to better target COVID-19 treatment. Meanwhile, some blanks in the study can also be solved in future research. First, it is necessary to determine the specific standards adopted in different conditions for the government's medical service model for the elderly population. Second, the results of the government's medical service model for the elderly population can be transformed into practical policy recommendations for reference in the severe epidemic areas. Third, in the process of medical service model in different areas, the government and the elderly population should determine the order of action of relevant research, rather than taking action at the same time.

## Supporting information

**S1 File.**
(DOCX)

## Author Contributions

**Conceptualization:** Yuntao Bai, Lan Wang.

**Data curation:** Yuntao Bai, Shuang Xu.

**Formal analysis:** Yuntao Bai, Lan Wang, Shuang Xu.

**Funding acquisition:** Lan Wang.

**Investigation:** Yuntao Bai.

**Methodology:** Yuntao Bai.

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
