## [Decision Letter · Decision Letter 0]

24 Oct 2023

PONE-D-23-25723Health improvement of the elderly in five Central Asian countries during COVID-19 based on difference gamePLOS ONE

Dear Dr. Wang,

Thank you for submitting your manuscript to PLOS ONE. After careful consideration, we feel that it has merit but does not fully meet PLOS ONE’s publication criteria as it currently stands. Therefore, we invite you to submit a revised version of the manuscript that addresses the points raised during the review process. Please follow both reviewers comments, which I found useful and agree with. Moreover, please improve the literature review. In particular, check the works that specifically studied Central Asian Republics during COVID-19, e.g.: - Alfano, V., Ercolano, S., & Pinto, M. (2023). Modeling Central Asia’s Choices in Containing COVID-19: A Multivariate Study. *Administration & Society*, *55*(9), 1819-1836, that addressed the differences and similarities among Central Asian countries during the first pandemic wave;- Alfano V. COVID-19 in Central Asia: exploring the relationship between governance and non-pharmaceutical intervention. Health Policy Plan. 2022 Sep 13;37(8):952-962, that studied the impact of governance on the evolution of the pandemic in Central Asian countries.

We look forward to receiving your revised manuscript.

Kind regards,

Vincenzo Alfano

Academic Editor

PLOS ONE

5. Please remove your figures from within your manuscript file, leaving only the individual TIFF/EPS image files, uploaded separately. These will be automatically included in the reviewers’ PDF.

6. We notice that your supplementary figures (Appendix 1-3) are included in the manuscript file. Please remove them and upload them with the file type 'Supporting Information'. Please ensure that each Supporting Information file has a legend listed in the manuscript after the references list.

Reviewers' comments:

Reviewer's Responses to Questions

**Comments to the Author**

1. Is the manuscript technically sound, and do the data support the conclusions?

Reviewer #1: Yes

Reviewer #2: Yes

2. Has the statistical analysis been performed appropriately and rigorously? 

Reviewer #1: I Don't Know

Reviewer #2: I Don't Know

3. Have the authors made all data underlying the findings in their manuscript fully available?

Reviewer #1: Yes

Reviewer #2: Yes

4. Is the manuscript presented in an intelligible fashion and written in standard English?

Reviewer #1: Yes

Reviewer #2: Yes

5. Review Comments to the Author

Reviewer #1: Thank you for letting me review this paper! First of all, I have to disclose that I am no expert in the game theory model. I have tried my best to assess the paper and to integrate the knowledge I have in my area of expertise. But if you find that any comment re: game theory is rubbish, please feel absolutely free to say so.

I will outline the points that came to my attention under the resp. headlines.

Abstract:

It would be useful if you could briefly describe in the abstract what is understood by "the green channel", thus one can better assess the results that you outline.

1. Introduction:

1.1 Background and research significance:

- I would deem it worthwhile if you included more sources, esp. in the first part where you report the numbers (e.g., WHO sources), bt, importantly, also when you argue that "make some administrative changes to their existing healthcare systems" as this seems to be the starting point for the main argument of your paper.

- Please explain "the hospital opens a "green channel" for the elderly" - what exactly does this green channel include? It doesn't get quite clear here either.

1.2 Literature:

- You write "Among them, the physical health of the elderly is affected more." To what is this compared? People with existing health issues are also very much affected (compared to those with no existing health issues). Please clarify the comparative figure and indicate a source. Thus, you could condense the following part where you outline who has studied what to a concise argument.

- The literature part somewhat looks more like a list of literature evidence than a chain of reasoning. This part could be more strongly summarized/condensed.

-The "game" part (parties, theory), on the other hand, is only introduced in the method section. I feel it would be more stringent to bring this part forward in chapter 1.

2. Methodology

2.1.1 Problem description

- Here again, you describe the different methods of caretaking (Green channel etc.). If you put this forward in section 1, you take the reader along from the beginning and strengthen the paper's line of argumentation (and save some characters/reading).

As I'm not familiar with in the method/ game model, I will refrain from assessing the rest of chapter 2.

3. Results

- I like te arguing from proposition to conclusion which makes it easy(ier) for the readers to follow even without fully grasping the calculations.

- However, you argue a lot that X enhances or reduces the credibility of the government. Maybe introduce this variable (credibility of the government) already at an earlier stage of the paper so that it becomes clear from the beginning that this is an important part of your line of argumentation.

4. Discussion

This part is very well written. I just want to ask the authors to consider to integrate the discussion in the results section. (If you don't find this useful/doable, no problem, it is just what I find useful, esp. when you have a stand-alone chapter on conclusions.)

5. Conclusions

- Please discuss what we can learn from your research on Covid19 for other diseases/similar problems that might arise in the future. (You could e.g. further outline the last part on "reference significance for how to effectively reform the medical system of the US government and how to regulate generic drugs in India".)

- Please also discuss the limitations of your paper.

Reviewer #2: The statements seem to provide a foundation for research hypotheses related to COVID-19 management in Central Asian countries, particularly concerning the elderly population and healthcare infrastructure. To formulate specific hypotheses, you would need to state the research questions you want to address and the relationships you wish to investigate. Additionally, these hypotheses should be framed more precisely, with clear independent and dependent variables, to make them suitable for empirical testing.In general, the conclusions appear to be logically derived from the information presented in the text, with a focus on the dynamic nature of healthcare resource allocation during the pandemic.

6. PLOS authors have the option to publish the peer review history of their article (what does this mean?). If published, this will include your full peer review and any attached files.

Reviewer #1: No

Reviewer #2: No

---

## [Author Response · Author response to Decision Letter 0]

1 Nov 2023

Response to reviewer1

Dear Editors and Reviewers:

Many thanks for your valuable comments and suggestions on our manuscript entitled “Health improvement of the elderly in five Central Asian countries during COVID-19 based on difference game” (Manuscript ID: PONE-D-23-25723). The comments and suggestions are very helpful for improving our paper. We have made revision based on the comments and suggestions. Please find our response as follows, and we have made revision which marked in blue in the paper. Attached please find the revised version, which we would like to submit for your kind consideration.

Point 1：

Abstract:

It would be useful if you could briefly describe in the abstract what is understood by "the green channel", thus one can better assess the results that you outline.

Response 1: 

Thank you very much for your suggestion. In the revised version, this article explains "the green channel" in the abstract section, which is detailed in blue on lines 12-14.

Point 2：

 1. Introduction:

1.1 Background and research significance:

- I would deem it worthwhile if you included more sources, esp. in the first part where you report the numbers (e.g., WHO sources), bt, importantly, also when you argue that "make some administrative changes to their existing healthcare systems" as this seems to be the starting point for the main argument of your paper.

- Please explain "the hospital opens a "green channel" for the elderly" - what exactly does this green channel include? It doesn't get quite clear here either.

Response 2: 

Thank you very much for your suggestion. In the revised version, the sources for the first part of the reported figures are identified, which is detailed in blue on lines 33. At the same time, the source of the sentence "make some administrative changes to their existing healthcare systems" is also noted, which is detailed in blue on lines 48.

In the revised version, this article explains that the hospital opens a "green channel" for the elderly, and clarifies what this "green channel" is and what it includes. For details, see lines 57-64 in blue.

Point 3：

1.2 Literature:

- You write "Among them, the physical health of the elderly is affected more." To what is this compared? People with existing health issues are also very much affected (compared to those with no existing health issues). Please clarify the comparative figure and indicate a source. Thus, you could condense the following part where you outline who has studied what to a concise argument.

- The literature part somewhat looks more like a list of literature evidence than a chain of reasoning. This part could be more strongly summarized/condensed.

-The "game" part (parties, theory), on the other hand, is only introduced in the method section. I feel it would be more stringent to bring this part forward in chapter 1. 

Response 3: 

Thank you very much for your suggestion. "Among them, the physical health of the elderly is affected more." This is mainly compared to younger people. In the revised draft, this paper lists specific data to illustrate this point, and indicates the source of the data. For details, see lines 69-74 in blue. Through these data, this further illustrates the need to study the elderly population. Meanwhile, in the revised version, this paper has condensed the following section, which Outlines who studied what, into a concise argument, and has added some references along the way. For details, see lines 75-81 in blue.

In the revised version, in order to make the literature review part more like an inference chain, the literature review part can be summarized/condensed more effectively in this paper. For details, see lines 75-81 and 84-87 in blue.

In the revised version, the "Game" part (Parties, theories) is presented in the first chapter and is introduced in detail. For details, see lines 105-109 in blue.

Point 4：

2. Methodology

2.1.1 Problem description

- Here again, you describe the different methods of caretaking (Green channel etc.). If you put this forward in section 1, you take the reader along from the beginning and strengthen the paper's line of argumentation (and save some characters/reading).

Response 4: 

Thank you very much for your suggestion. In the revised version, this point(Green channel etc.) is made in Part 1, which strengthens the argument line of the paper. For details, see lines 57-64 in blue.

Point 5：

3. Results

- I like te arguing from proposition to conclusion which makes it easy(ier) for the readers to follow even without fully grasping the calculations.

- However, you argue a lot that X enhances or reduces the credibility of the government. Maybe introduce this variable (credibility of the government) already at an earlier stage of the paper so that it becomes clear from the beginning that this is an important part of your line of argumentation.

Response 5: 

Thank you very much for your suggestion. This variable was introduced earlier in this article. However, in order to make the research method of the paper more clear, in the revised version, the utility function and state variable X are introduced in detail. For details, see lines 273-292 in blue.

Point 6：

5. Conclusions

- Please discuss what we can learn from your research on Covid19 for other diseases/similar problems that might arise in the future. (You could e.g. further outline the last part on "reference significance for how to effectively reform the medical system of the US government and how to regulate generic drugs in India".)

- Please also discuss the limitations of your paper.

Response 6: 

Thank you very much for your suggestion. In the modified version, this article further summarizes the last part. "reference significance for how to effectively reform the medical system of the US government and how to regulate generic drugs in India". For details, see lines 593-596 in blue. Meanwhile, this sentence is also explained and discussed in the discussion section. For details, see lines 549-572 in blue. 

Response to reviewer2

Dear Editors and Reviewers:

Many thanks for your valuable comments and suggestions on our manuscript entitled “Health improvement of the elderly in five Central Asian countries during COVID-19 based on difference game” (Manuscript ID: PONE-D-23-25723). The comments and suggestions are very helpful for improving our paper. We have made revision based on the comments and suggestions. Please find our response as follows, and we have made revision which marked in blue in the paper. Attached please find the revised version, which we would like to submit for your kind consideration.

Point ：

The statements seem to provide a foundation for research hypotheses related to COVID-19 management in Central Asian countries, particularly concerning the elderly population and healthcare infrastructure. To formulate specific hypotheses, you would need to state the research questions you want to address and the relationships you wish to investigate. Additionally, these hypotheses should be framed more precisely, with clear independent and dependent variables, to make them suitable for empirical testing.In general, the conclusions appear to be logically derived from the information presented in the text, with a focus on the dynamic nature of healthcare resource allocation during the pandemic.

Response : 

Thank you very much for your suggestion. In the revised version, in order to form specific hypotheses, this paper states the research problem that is intended to be solved. For details, see lines 226-231 in blue. In a modified version, this article states the relationship you want to investigate in order to form a specific hypothesis. For details, see lines 208-225 in blue. 

In the revised version, in order that the framework of these assumptions should be more precise, the independent and dependent variables are stated in the assumptions. For details, see lines 239-240, 249-250, 253-255 and 257-258 in blue.

In the revised version, this paper describes the dynamic nature of health care resource allocation during a pandemic. For details, see lines 138-168 in blue.

---

## [Decision Letter · Decision Letter 1]

7 Nov 2023

Health improvement of the elderly in five Central Asian countries during COVID-19 based on difference game

PONE-D-23-25723R1

Dear Dr. Wang,

We’re pleased to inform you that your manuscript has been judged scientifically suitable for publication and will be formally accepted for publication once it meets all outstanding technical requirements.

Kind regards,

Vincenzo Alfano

Academic Editor

PLOS ONE

Additional Editor Comments (optional):

Reviewers' comments:

Reviewer's Responses to Questions

**Comments to the Author**

1. If the authors have adequately addressed your comments raised in a previous round of review and you feel that this manuscript is now acceptable for publication, you may indicate that here to bypass the “Comments to the Author” section, enter your conflict of interest statement in the “Confidential to Editor” section, and submit your "Accept" recommendation.

Reviewer #1: All comments have been addressed

Reviewer #2: All comments have been addressed

2. Is the manuscript technically sound, and do the data support the conclusions?

Reviewer #1: Yes

Reviewer #2: Yes

3. Has the statistical analysis been performed appropriately and rigorously? 

Reviewer #1: I Don't Know

Reviewer #2: Yes

4. Have the authors made all data underlying the findings in their manuscript fully available?

Reviewer #1: Yes

Reviewer #2: Yes

5. Is the manuscript presented in an intelligible fashion and written in standard English?

Reviewer #1: Yes

Reviewer #2: Yes

6. Review Comments to the Author

Reviewer #1: Thank you for addressing my points so thoroughly! Although there were no blue parts (at least not in my version), indicating the lines where you changed sth to the manuscript helped me figure out resp. changes.

Happy to read the final paper when it's published.

Reviewer #2: (No Response)

7. PLOS authors have the option to publish the peer review history of their article (what does this mean?). If published, this will include your full peer review and any attached files.

Reviewer #1: No

Reviewer #2: **Yes: **Manuel Gandoy-Crego

---

## [Editor Report · Acceptance letter]

17 Nov 2023

PONE-D-23-25723R1 

Health improvement of the elderly in five Central Asian countries during COVID-19 based on difference game 

Dear Dr. Wang:

I'm pleased to inform you that your manuscript has been deemed suitable for publication in PLOS ONE. Congratulations! Your manuscript is now with our production department. 

Kind regards, 

on behalf of

Dr. Vincenzo Alfano 

Academic Editor

PLOS ONE